# Newborn Screening for Gaucher Disease: Parental Stress and Psychological Burden

**DOI:** 10.3390/ijns11010014

**Published:** 2025-02-14

**Authors:** Chiara Cazzorla, Vincenza Gragnaniello, Giacomo Gaiga, Daniela Gueraldi, Andrea Puma, Christian Loro, Giada Benetti, Rossana Schiavo, Elena Porcù, Alessandro P. Burlina, Alberto B. Burlina

**Affiliations:** 1Division of Inherited Metabolic Diseases, Department of Women’s and Children’s Health, University Hospital of Padua, 35128 Padua, Italy; chiara.cazzorla@aopd.veneto.it (C.C.); vincenza.gragnaniello@aopd.veneto.it (V.G.); giacomo.gaiga@aopd.veneto.it (G.G.); daniela.gueraldi@aopd.veneto.it (D.G.); andrea.puma@aopd.veneto.it (A.P.); christian.loro@aopd.veneto.it (C.L.); dietista.giadabenetti@gmail.com (G.B.); elena.porcu@aopd.veneto.it (E.P.); 2U.O.C. Hospital Psychology, University Hospital of Padua, 35128 Padua, Italy; rossana.schiavo@aopd.veneto.it; 3Neurology Unit, St Bassiano Hospital, 36061 Bassano del Grappa, Italy; alessandro.burlina@aulss7.veneto.it

**Keywords:** lysosomal newborn screening, Gaucher disease, parenting stress index—short form, parental stress

## Abstract

In the last few decades, neonatal screening (NBS) has expanded to include lysosomal storage diseases, allowing for the early identification of both symptomatic and asymptomatic cases. However, neonatal diagnosis of late-onset disorders can cause parental stress and affect family well-being, possibly leading to overmedicalization. The impact of a positive NBS for Gaucher disease type 1 (GD1) can have an important impact on parental psychological well-being and psychosocial functioning. This study aims to study parental stress in parents of newborns who had a positive result for Gaucher disease in an NBS program in Northeastern Italy. Fourteen parents (7 fathers and 7 mothers) of seven children with confirmed GD1 (86% boys) completed the Parenting Stress Index—Short Form (PSI-SF) at diagnosis (T0), 12 months (T1), and 36 months (T2). A control group of fourteen parents (7 fathers and 7 mothers) whose children had normal NBS results was included. Interviews were conducted for the GD1 group at T2 to investigate the usefulness of the NBS program. At T0, higher parental stress was assessed in GD1 parents compared to the healthy controls. Subsequently, the parents of GD1 children reported significant reductions in Parental Distress at T1 compared to T0. Mothers showed further reductions at T2, while the fathers’ distress decreased but not significantly. GD1 mothers had significantly higher distress scores than the controls at T1, but this difference diminished over time. Our study highlights the psychological impact of NBS on GD1, emphasizing the need for better multidisciplinary communication to reduce parental stress throughout the diagnostic and treatment process.

## 1. Introduction

Newborn Screening (NBS) for Inborn Errors of Metabolism has gained increasing importance due to the development of high-throughput assays and novel therapies, significantly improving outcomes and quality of life for children and their parents [1,2]. In many developed countries, mass spectrometry (MS/MS) technology has been implemented to detect up to 40–50 severe diseases, including lysosomal storage diseases (LSDs) [3].

Nevertheless, the selection of disease panels for expanded NBS remains debated, with varying inclusion criteria across different countries. In the United States, some lysosomal disorders (Pompe disease and mucopolysaccharidosis type I and II) have been included in the Recommended Universal Screening Panel (RUSP), but Gaucher disease is currently not included.

In Italy, NBS became mandatory in 2016, and the current panel includes almost 40 metabolic defects (amino acid, urea cycle, organic acid, and fatty acid β-oxidation disorders). Since 2015, the regional Health Government of Northeastern Italy has expanded its NBS program to include four lysosomal storage diseases: Gaucher disease, Fabry disease, Pompe disease, and mucopolysaccharidosis type I [2]. Among these, Gaucher disease has a relatively high incidence in Italy [4], imposing a significant healthcare burden.

To date, little is known about the social and psychological effects of NBS programs.

For this reason, it is crucial to examine the impact of NBS result communication on parental stress [5], considering that receiving an abnormal result can be perceived as life-altering news.

The manner in which the diagnosis is communicated influences not only short-term parental outcomes but also long-term effects on both parents and children [6,7].

An Italian study indicated that parents of 32 infants who received positive results from expanded NBS reported experiencing discomfort potentially related to the communication of the results [8].

Most studies assessing the impact of NBS communication have focused on cystic fibrosis and sickle cell disease. A recent systematic review analyzing the psychosocial issues related to NBS reported that parents of infants who tested positive for metabolic disorders experienced short-term anxiety and distress [9]. Moreover, a recent longitudinal questionnaire-based study reported that parents and children receiving a positive NBS result may experience important psychosocial and financial burdens, with a greater impact on caregivers in the younger ages of their children and with a risk on long-term effects on individual health [10].

Other studies have examined stress levels in parents of children diagnosed with genetic metabolic disorders, showing that mothers of children identified through NBS reported significantly lower Parenting Stress Index (PSI) scores compared to mothers whose children were identified clinically [11]. The waiting period and the delays for confirmatory diagnostic testing were particularly distressing for parents, with some reporting clinical levels of depression during this time [12]. Moreover, a recent study investigating parents of children with Phenylketonuria (PKU), a metabolic disorder, revealed that mothers and fathers often report similar challenges in managing their child’s daily needs; this may significantly impact the child’s psychological well-being [13].

These considerations are fundamental when considering NBS for rare diseases, such as lysosomal diseases, which have variable phenotypes and late-onset presentations.

Nevertheless, for most lysosomal storage disorders, NBS detects newborns across the full phenotypic spectrum of severity, but not all subtypes benefit equally from early detection [14]. Furthermore, the variability in age of onset can create uncertainty, leading to the phenomenon of “patients in waiting” [14,15,16], a condition in which living between sickness and health, potentially imposing a psychological burden on families. Despite the existing knowledge, few studies have evaluated parental stress in the long term.

Therefore, we chose to evaluate parental stress in families of neonates who screened positive for Gaucher disease through the NBS program in Northeastern Italy.

The analysis was conducted at three time points, at diagnosis confirmation, one year later, and three years later, with the aim of understanding both short-term and long-term parental outcomes.

## 2. Materials and Methods

### 2.1. Study Design

From Semptember 2015 to Semptember 2024, a total of 275,011 newborns were screened at the Reference Centre for Expanded Newborn Screening, Division of Inherited Metabolic Diseases, at the University Hospital of Padua (Italy).

Of these, 16 newborns tested positive for Gaucher disease (GD). Molecular analysis confirmed that 14 cases were type 1 Gaucher disease (GD1). Communication of the diagnosis occurs with the confirmation of a pathologic genetic test. This happens when the newborn is 1–2 months old.

A multidisciplinary team, including a metabolic specialist, a geneticist, and a psychologist, was present during the communication of the diagnosis to the parents.

For this study, eligible participants included parents of children diagnosed with GD1. Of the 14 patients confirmed with GD1, four were excluded due to a linguistic barrier, and three others were excluded due to the parents’ refusal to participate.

All children of our sample diagnosed with GD1 were asymptomatic when the study started and had not yet initiated therapy; all remained asymptomatic for the entire study.

In addition, we recruited an equal number of parents of children with negative newborn screening results as a control group.

The parents of children diagnosed with GD1 completed the PSI-SF, a widely used instrument to assess the level of stress in the parent–child relationship, along with a brief structured interview. For the GD1 group, the assessment was administered as follows:At the time of diagnosis confirmation, when the child was approximately 1–2 months old (T0), parents completed the PSI-SF;When the child was around 12 months old (T1), parents completed the PSI-SF and participated in the short structured interview;When the child reached approximately 36 months of age (T2), parents again completed the PSI-SF and the short, structured interview.

Regarding the control group, the parents of children with negative NBS results similarly completed the same PSI-SF questionnaire when their children were approximately 1–2 months old (T0), 12 months old (T1), and 36 months old (T2).

A psychologist was available during the completion of the questionnaires in case of any question about the compilation and also conducted the interview. Written informed consent was obtained prior to the interview.

This study was approved by the Research Ethics Committee of the University Hospital of Padua (n. 39776; 5 June 2024).

### 2.2. Questionnaires

#### 2.2.1. Parenting Stress Index—Short Form

The Parenting Stress Index—Short Form (PSI-SF) [17] is a 36-item self-administered questionnaire that provides a Total Stress score and three subscale scores:

Parental Distress, Parent–Child Dysfunctional Interaction, and Difficult Child. The Total Stress score is a measure of the overall level of stress related to the parental role. The Parental Distress (PD) subscale measures the extent to which parents feel competent, restricted, conflicted, supported, and/or depressed in their role as a parent. The Parent–Child Dysfunctional Interaction (P-CDI) subscale measures the extent to which parents feel satisfied with their child and their interactions with them. The Difficult Child (DC) subscale assesses how a parent perceives their child, particularly whether the child is easy or difficult to take care of.

Higher total and subscale scores reflect greater parenting stress. The normal range for Total Stress scores is 55 to 85, with scores above 85 considered to be in the clinical range. A Total Stress score above 90 indicates that a parent is experiencing clinically significant levels of stress.

The PSI-SF also provides a Defensive Responding Index, which serves as an internal validity check based on the parents’ responses. For this index, scores of ≤10 suggest that the validity of the Total Stress and subscale scores may be questionable.

The test–retest reliability for the PSI-SF is high, and both concurrent and predictive validity have been demonstrated for the full-length version of the questionnaire. The short form correlates very highly with the full-length version [15]. This questionnaire was used in its Italian version [18]. The PSI-SF was administered to both mothers and fathers separately.

#### 2.2.2. Short, Structured Interview

The parents of children diagnosed with Gaucher disease completed a short, structured interview consisting of three dichotomous questions (yes/no responses) related to their role in their child’s care, their understanding of the newborn screening (NBS) test, and their perceptions of the usefulness of screening.

The following questions or sentences were administered separately to both mothers and fathers:*My partner can’t understand my point of view regarding our baby’s management.**Do you feel that you received and understood all the useful information relating to Newborn Screening?**Would you have preferred not to know about your child’s disease considering that the therapy will be start only with the onset of clinical manifestation?*

### 2.3. Statistical Analyses

A two-tailed *t*-test was used for statistical comparisons in the data analysis. The following specific comparisons were made:

Within the GD1 group:Comparison of PSI-SF scores at the time of diagnosis confirmation (T0, when the child was 1–2 months old) vs. PSI-SF scores at 12 months (T1);Comparison of PSI-SF scores at the time of diagnosis confirmation (T0) vs. PSI-SF scores at 36 months (T2).

Between the GD1 group and the control group:Comparison of PSI-SF scores at 1–2 months (T0) in the GD1 group vs. the control group;Comparison of PSI-SF scores at 12 months (T1) in the GD1 group vs. the control group;Comparison of PSI-SF scores at 36 months (T2) in the GD1 group vs. the control group.

These comparisons were aimed at assessing how stress levels in the parent–child relationship, as measured by the PSI-SF, evolved over time within the GD1 group, and how they differed between parents of children with GD1 and those in the control group, who had negative newborn screening (NBS) results. Statistical analyses were performed using the JASP software (version 0.17.1) [19], with the significance threshold set at *p* < 0.05. The PSI-SF scores for mothers and fathers were analyzed separately to investigate potential differences between parental roles in managing stress related to their child’s diagnosis and development.

For the structured interview, responses were reported in percentage form, summarizing the distribution of answers to capture trends and common themes among the parents’ experiences and perspectives.

## 3. Results

### 3.1. Study Cohort

A total of 28 parents (14 mothers and 14 fathers) were enrolled. The sample consisted of 14 parents (7 mothers and 7 fathers) of children diagnosed with GD1 and 14 parents (7 mothers and 7 fathers) of children with negative newborn screening results. All the GD1 and control newborns considered in this study were full-term, with a weight comparable to healthy peers of the same age. No infant, at the time of birth or during the 36-month follow-up showed any clinical signs of the disease. Psychomotor development, assessed with a standardized developmental scale, was normal for all GD1 children.

Outpatient visits were performed every 6 months during the first year of life and annually in the following years. The follow-up visits included biochemical tests, abdominal ultrasounds, and specific assessments to define and monitor the acquisition of psychomotor development milestones over time through the administration of age-appropriate psychometric and developmental scales.

We collected and described demographic information, such as age, ethnicity, educational level, marital status, and employment status; the presence of other children in both groups; gender, genetic, and molecular analyses; and biochemical data, in the GD1 population (Table 1 and Table 2, respectively).

The groups were similar in terms of age, ethnicity, marital status, and professional status. Parents in the control group presented higher educational levels, with 79% having higher education compared to 21% in the GD1 group.

### 3.2. Parental Stress (PSI-SF)

Table 3 and Table 4 report the mean percentile scores and standard deviations for the GD1 group at T0 (diagnosis confirmation) versus T1 (12 months) and T0 versus T2 (36 months), respectively. These tables show a general decrease in PSI-SF scores for most scales for both mothers and fathers. This reduction in scores suggests a tendency for perceived parental stress to decrease over time, particularly in the long term.

Table 5, Table 6 and Table 7 present a comparison of PSI-SF scores between the GD1 group and the control group (with negative NBS) at T0, T1, and T2, respectively.

#### 3.2.1. Comparison of PSI-SF Results at Confirmation of Diagnosis (T0) vs. PSI-SF Results at 12 Months (T1) in GD Group

In the GD1 group, there was a significant reduction in scores on the Parental Distress Scale at T1 compared to T0 for both mothers and fathers (*p* = 0.03 and *p* = 0.04, respectively). This significant decrease suggests that perceived parental stress tends to decrease within the first 12 months after diagnosis.

The remaining scales decrease but not significantly (Table 3).

**Table 3 IJNS-11-00014-t003:** Comparison between mean (DS) PSI-SF percentile scores at confirmation of diagnosis (T0) and at 12 months (T1) in GD1 group (* *p*-value < 0.05).

	Gaucher Group			
Parenting Stress Index–SF	Mothers		Fathers	
	Mean (ds)	*p*-Value	Mean (ds)	*p*-Value
PSI-SF Total Score				
T0	56 (20)	0.69	49 (22)	0.11
T1	51 (23)		29 (26)	
PSI-SF Parental Distress				
T0	49 (26)	0.03 *	38 (21)	0.03 *
T1	24 (9)		18 (23)	
PSI-SF Parent–Child Dysf. Int.				
T0	68 (16)	0.17	48 (16)	0.23
T1	51 (17)		33 (23)	
PSI-SF Difficult Child				
T0	49 (12)	0.08	63 (22)	0.21
T1	71 (26)		48 (25)	

#### 3.2.2. Comparison of PSI-SF Results at Confirmation of Diagnosis (T0) vs. PSI-SF Results at 36 Months (T2) in GD Group

The GD1 group showed a further decline in stress scores, indicating that parental stress tends to decrease over time, at T2, when the children reached 36 months of age.

This reduction was significant among mothers in the Parental Distress Scale (*p* = 0.02), the Parent–Child Dysfunctional Interaction Scale (*p* = 0.02), and the Total Stress Scale (*p* = 0.04). These results suggest that perceived stress significantly decreased for mothers over the 36-month period (Table 4).

**Table 4 IJNS-11-00014-t004:** Comparison between mean (DS) PSI-SF percentile scores at confirmation diagnosis (T0) and 36 months (T2) in GD1 group (* *p*-value < 0.05).

	Gaucher Group			
Parenting Stress Index–SF	Mothers		Fathers	
	Mean (ds)	*p*-Value	Mean (ds)	*p*-Value
PSI-SF Total Score				
T0	56 (20)	0.04 *	49 (22)	0.09
T2	31 (32)		27 (15)	
PSI-SF Parental Distress				
T0	49 (26)	0.02 *	38 (21)	0.17
T2	20 (27)		22 (19)	
PSI-SF Parent–Child Dysf. Int.				
T0	68 (16)	0.02 *	48 (16)	0.07
T2	31 (22)		26 (12)	
PSI-SF Difficult Child				
T0	49 (12)	0.87	63 (22)	0.68
T2	46 (35)		57 (22)	

#### 3.2.3. Comparison of GD1 Group’s PSI-SF Results vs. Control Group’s PSI-SF Results at 1–2 Months (T0), 12 Months (T1), and at 36 Months (T2)

At T0, the mothers of children with GD1 reported significantly higher scores compared to the mothers of children in the control group in the Total Stress Score (*p* = 0.01), and the Parent–Child Dysfunctional Interaction Scale (*p* < 0.01). No significant differences were observed between the fathers of children with GD1 and in the control group (Table 5).

**Table 5 IJNS-11-00014-t005:** Comparison between Gaucher and negative screening results at T0. Mean and standard deviation of the percentiles obtained from the raw scores (* *p*-value < 0.05).

	Gaucher Group vs. Control Group			
Parenting Stress Index–SF	Mothers		Fathers	
	Mean (ds)	*p*-Value	Mean (ds)	*p*-Value
PSI-SF Total Score				
Gaucher	56 (20)	0.01 *	49 (22)	0.08
Control	25 (18)		26 (22)	
PSI-SF Parental Distress				
Gaucher	49 (26)	0.14	38 (21)	0.12
Control	27 (25)		19 (24)	
PSI-SF Parent–Child Dysf. Int.				
Gaucher	68 (16)	<0.01 *	48 (16)	0.13
Control	28 (9)		35 (13)	
PSI-SF Difficult Child				
Gaucher	49 (12)	0.06	63 (22)	0.19
Control	35 (12)		44 (29)	

At T1, similarly to T0, the mothers of children with GD1 reported significantly higher scores compared to the mothers of children in the control group in the Total Stress Score (*p* = 0.02), in the Parent–Child Dysfunctional Interaction Scale (*p* = 0.03), and, additionally, in the Difficult Child Scale (*p* = 0.04). Also, at T1, no significant differences were observed between the fathers of children with GD1 and the fathers of the control group (Table 6).

**Table 6 IJNS-11-00014-t006:** Comparison between Gaucher and negative screening results after 12 months (T1). Mean and standard deviation of the percentiles obtained from the raw scores (* *p*-value < 0.05).

	Gaucher Group vs. Control Group			
Parenting Stress Index–SF	Mothers		Fathers	
	Mean (ds)	*p*-Value	Mean (ds)	*p*-Value
PSI-SF Total Score				
Gaucher	51 (23)	0.02 *	29 (26)	0.85
Control	24 (18)		26 (22)	
PSI-SF Parental Distress				
Gaucher	24 (9)	0.75	18 (23)	0.89
Control	27 (25)		19 (24)	
PSI-SF Parent–Child Dysf. Int.				
Gaucher	51 (17)	0.03 *	33 (23)	0.82
Control	29 (8)		35 (13)	
PSI-SF Difficult Child				
Gaucher	71 (26)	0.04 *	48 (25)	0.83
Control	38 (17)		44 (29)	

At T2, no statistically significant differences emerged between the mothers of GD1 children and the mothers in the control group. However, in a comparison of the fathers, a statistically significant difference emerged in the Difficult Child Scale, with the fathers of children with GD1 reporting higher scores than the fathers in the control group (*p* = 0.03) (Table 7).

**Table 7 IJNS-11-00014-t007:** Comparison between Gaucher and negative screening results after 36 months (T2). Mean and standard deviation of the percentiles obtained from the raw scores (* *p*-value < 0.05).

	Gaucher Group vs. Control Group			
Parenting Stress Index–SF	Mothers		Fathers	
	Mean (ds)	*p*-Value	Mean (ds)	*p*-Value
PSI-SF Total Score				
Gaucher	31 (32)	0.84	27 (15)	0.95
Control	28 (22)		26 (22)	
PSI-SF Parental Distress				
Gaucher	20 (27)	0.36	22 (19)	0.83
Control	33 (27)		24 (27)	
PSI-SF Parent–Child Dysf. Int.				
Gaucher	31 (22)	1.00	26 (12)	0.62
Control	31 (19)		30 (19)	
PSI-SF Difficult Child				
Gaucher	46 (35)	0.58	57 (22)	0.02 *
Control	37 (25)		26 (21)	

### 3.3. Short, Structured Interview

The mothers and fathers of children with GD1 did not report any conflicts regarding the management of their child’s condition. With regard to the knowledge of newborn screening, three mothers and one father of the GD1 group (4 out of 14 parents; 29%) reported that they did not understand all the information related to the Newborn Screening test. The remaining 10 parents (71%) reported that they had received and understood all the necessary information about NBS. When asked about the usefulness of the screening (“Would you have preferred not to know about your child’s disease considering that the therapy will be start only with the onset of clinical manifestation?), two mothers and one father (3 out of 14 parents; 21%) reported that they would have preferred not to know about their child’s disease. However, the majority (11 out of 14 parents; 79%) felt that the newborn screening results were valuable in helping them understand the condition.

These results remained stable at T2 (36 months). Additionally, no differences in responses were found based on the parents’ social status.

## 4. Discussion

The goal of NBS is to detect potentially fatal or serious treatable conditions in newborns as early as possible, before the onset of symptoms. Early detection enables timely intervention and initiation of treatment [14]. The new NBS programs have been expanded to include acute metabolic disorders, such as severe forms of beta-oxidation deficiency and organic acidurias. While the value of screening for certain acute metabolic conditions is widely recognized, the identification of diseases with a non-acute neonatal onset presents unresolved challenges. One of these challenges arises from the absence of acute symptoms, which can make it more difficult for parents to accept the diagnosis.

Indeed, the management of asymptomatic individuals and the heightened state of alertness of the parents are important and complex problems. More than two decades after the introduction of MS/MS-based neonatal screening programs, these challenges have been found to contribute to the absence of a uniform screening panel and a lack of consensus on the criteria for disease inclusion [20].

Furthermore, NBS programs must consider the need to minimize potential harms to children, parents, and families [21].

The communication of diagnoses detected through NBS for chronic diseases has a significant impact on parental and family distress [5].

Caregivers of ill children often experience difficulties in dealing with stress and emotional challenges, as well as adapting to new responsibilities, treatment regimens, and the overall reduction in one’s quality of life [22]. Stress and coping strategies can significantly influence the quality of life of both patients and caregivers [23].

This study presents data from the experience in Northeastern Italy, assessing the effects of positive results of newborn screening for Gaucher disease on parental stress at the Center for Newborn Screening within the Division of Inherited Metabolic Diseases in Padova (Italy). Gaucher disease type 1 was selected for analysis due to the ongoing controversy surrounding its inclusion in newborn screening programs.

Neonates with GD1 detected through NBS receive the diagnosis shortly after birth, but treatment is not initiated until the onset of symptoms. The results from our study highlight that parents of newborns with positive screening results report a decrease in psychological distress following the confirmation of the diagnosis, with reductions in stress observed over time for both mothers and fathers. Furthermore, our study indicates a significant long-term decline in parental stress, probably due to a process of adaptation to the disease, during which parents develop coping strategies to manage their child’s condition in everyday life.

Interestingly, our study shows that some aspects of parental stress perception are comparable between parents of children with GD1 and control group parents, even at the time of diagnosis confirmation. This is particularly evident in the perceptions of fathers. This could be partially explained by improved health outcomes in those children diagnosed through NBS, who require low rates of hospitalization and the absence of acute, life-threatening symptoms at onset. Our results underline that the time elapsed from communication of the diagnosis is an element that supports the activation of an adaptation process to the new disease condition, a new reality that parents must navigate. When provided with adequate support, parents can acquire and implement coping strategies that progressively reduce their perceived stress over time.

Some authors have noted that parents of children affected by lysosomal disorders, such as Fabry disease; Gaucher disease; mucopolysaccharidosis type I; Pompe disease; Krabbe disease; and Niemann Pick A, A/B, and B, diagnosed through NBS, are less likely to experience depression or significant parental stress compared to parents of children diagnosed through clinical presentation. This may be due, in part, to the nature of lysosomal disorders compared to other conditions identified through newborn screening [24].

Further, four parents (29%) reported that they did not understand all the information relating to NBS communication. Notably, among these parents, a couple had a low level of education; the same couple expressed a preference for not knowing about their child’s disease. Additionally, a smaller but important number of parents of children with GD1 (21%) reported that they would have preferred not to know about their child’s disease.

The low level of understanding regarding NBS among parents suggests that the information provided is either inadequate or not comprehended, as highlighted in other research.

These findings emphasize the importance of effective and empathetic communication to address parents’ needs and concerns, as well as to offer psychological support.

A systematic review analyzing psychosocial issues related to NBS indicates that the impact of unexpected communications can be mitigated through improved education about NBS, effective communication strategies, individualized genetic counseling, and anticipatory developmental guidance [9]. Based on these results, we can reasonably confirm that effective communication can promote the achievement of adequate resilience and more effective resources for managing their child’s illness, even in the long term.

Moreover, our results highlight the importance of considering both mothers’ and fathers’ perspectives. In our experience, fathers sometimes decline to attend outpatient visits, while mothers tend to feel a greater sense of responsibility.

Our experience suggests that the differences in scores obtained by mothers and fathers may be related to mothers assuming more of the caregiver role for their sick children. Qualitatively, our analysis revealed no differences based on social status or cultural characteristics within the sample.

Many studies reported that positive neonatal screening results entail psychological risks. Specifically, abnormal NBS results requiring retesting are associated with parental anxiety and/or depression, even when subsequent tests return normal [25]. Positive NBS results have been demonstrated to cause distress, alter behavior, and even to influence the development of new parental and family identities [26].

Particularly, NBS for lysosomal disorders raises several ethical and social issues that necessitate sensitive decision-making. Indeed, while newborn screening is more accepted for certain conditions (e.g., Pompe disease and mucopolysaccharidosis type I) because in the infantile forms, early diagnosis can alter the prognosis, the situation is more contentious for other diseases, such as Gaucher disease [27]. Newborns positive for GD1 are asymptomatic at birth, and the age at onset and clinical presentation of Type I GD is variable [28]. Furthermore, the onset is unpredictable and can potentially occur at any time from infancy to adulthood. Some authors support a watch-and-wait approach rather than an early start of GD1-specific therapy for asymptomatic or mildly symptomatic patients [29]. Therefore, it is desirable to think that treatment should begin upon the presentation of signs or symptoms related to the disease. Additionally, it is possible that some newborns with positive NBS results may never develop symptoms [29]. These considerations raise an important discussion regarding the ethical aspects related to the early identification of this disease through NBS. Therefore, diagnosis communication may be even more complex and underscores the importance of global and multidisciplinary care for the newborn and their family to support parents in maintaining a sufficiently good quality of life. Therefore, a comprehensive approach to care and support for these families is essential.

A recent qualitative study by Lisi and colleagues [30] reported that the majority of patients with lysosomal storage diseases that were interviewed were in favor of NBS.

Participants emphasized the positive aspects of early diagnosis, such as avoiding long diagnostic odysseys, feeling misunderstood, experiencing accusations of malingering, and ultimately desiring the ability to make different life decisions.

The impact of a diagnosis can dramatically alter the lives of patients’ families and caregivers. Providing information to parents and creating advocacy are crucial for the success of NBS programs, as this promotes a sense of trust and fosters support for NBS initiatives. Moreover, ensuring prompt response to abnormal results is essential for parents. Conversely, inadequate information can lead to a lack of trust in NBS and the healthcare system in general [31,32]. Moreover, it is essential to determine the appropriate timing for delivering information about NBS, as educating parents on the subject can enhance information retention [31] and, consequently, leads to better care for both children and parents.

Harrington et al. (2019) [33] developed an instrument to measure the impact on caregivers of caring for children with three lysosomal storage diseases: metachromatic leukodystrophy, mucopolysaccharidosis type I, and mucopolysaccharidosis type II. These authors identified several domains reported by caregivers that describe the impact of newborn screening: social and family life, emotional well-being, and physical and personal time. Social functioning included difficulties in spending time with family and friends and participating in leisure activities. Emotional impacts included feelings of helplessness and an inability to understand and communicate effectively with the child. Caregivers also reported limitations in daily activities, including challenges in making plans and completing work tasks. Additionally, the impact on physical health is manifested as feelings of physical exhaustion, sleep disturbances, and an inability to care for their own health.

Pruniski et al. (2017) [15], considering late-onset Pompe disease, one of the lysosomal storage diseases identified through NBS, highlighted that patients and families often find themselves in a state of being “patients in waiting”; these authors noted that living with uncertainty regarding diagnosis, treatment initiation, and the future of their children were predominant aspects of daily life. This psychological experience may be similar for parents of newborns with Gaucher disease.

In the case of Gaucher disease patients, a significant finding is related to diagnosis knowledge: frequently, these patients reported feeling relief, because they finally had a diagnosis, as it allows them to rule out more severe illnesses [34].

These findings highlight the importance of early communication of the diagnosis, facilitated by the NBS program. This opportunity can serve as a valuable resource in initiating a process of gaining awareness, which is crucial for both parents and patients in accepting the disease. With the NBS program, the figures involved in the communication of the diagnosis differ from those typically engaged when the patient is an adult. In the case of a positive NBS result, the patient is the newborn, but the decision-making responsibility is on the parents. Therefore, clinicians must consider the parents’ perspective until the young patient can actively participate in decisions.

Despite these findings, questions remain about the benefits of early diagnosis and the potential impact on family distress when a positive screening result is received.

Psychological support can play an essential role in the acceptance of chronic diseases, both for young patients and their families.

Considering the psychological impact of a disease diagnosis, it is crucial to establish a follow-up approach with parents of newborns affected by GD1 that is adequate not only for monitoring the onset of symptoms but also for providing parents with a space for listening and emotional processing. This is essential for building an appropriate therapeutic alliance. Our approach ensures that parents see the same professional figures (metabolic pediatricians and psychologists) at each follow-up visit, allowing them to benefit from continuity of care and appropriate support.

To our knowledge, previous studies have evaluated decision-making regarding NBS without considering parental stress, especially in the first year of the child.

## 5. Limitations

The small sample size and the variables related to sociocultural aspects and the region of origin of the parents may have affected the results. Further studies are needed to overcome these limitations and to produce more generalizable results.

Moreover, in our study we did not consider parents due to language barriers and parents who refused to participate, so these aspects could represent a selection bias. Future studies should aim to include all families, using instruments created in different languages.

An additional limitation is the disparity in educational levels between the control group, which showed a higher level of education, and the GD1 group. This discrepancy may have influenced the perception and acceptance of the NBS result and could be associated with stress levels at the time of diagnosis communication. Future research should account for one’s educational level as a factor that may affect the psychological burden experienced by parents of children with positive NBS results.

## 6. Conclusions and Future Directions

The NBS program for Gaucher disease raises several important questions, including the lack of homogeneity regarding the timing and the method of providing information about the screening process, as well as the timing for pharmacological therapies.

Although NBS is crucial for intervening before the onset of symptoms, thereby altering the prognosis of many screened diseases significantly, patients with GD1, considering that the age of onset can vary significantly and that some positive newborns may never develop symptoms, may perceive themselves as “patients in waiting.” The assessment of the impact of newborn screening on parental stress is an essential aspect for physicians and healthcare providers to gain a more comprehensive understanding of newborns with positive NBS results and their families. Further studies are needed to analyze long-term parental stress in metabolic diseases at risk of acute decompensation.

Studies with longer-term follow-ups are needed to evaluate how the adaptation process of parents of children with GD1 evolves over time. Moreover, future studies could focus on the experiences of fathers, along with mothers, to explore their perceptions of their roles and their psychological functioning in response to the communication of a child’s positive NBS result.

Establishing a standardized communication practice for newborn screening and implementing patient- and family-centered care is essential. This care model should include consistent and periodic contact with the metabolic team, even if infrequent controls are required. This approach could be important not only to guide and support parents but also to keep them informed about new therapeutic options for the treatment of Gaucher disease.

## Figures and Tables

**Table 1 IJNS-11-00014-t001:** Parents’ demographic characteristics.

	Positive-Screened Group Gaucher	Negative-Screened Group
	(n = 14)	(n = 14)
Age, years		
Range	23–43	32–42
Mean (DS)	36.8 (5.7)	36.9 (2.9)
Ethnicity, N (%)		
Caucasian	14/14 (100%)	14/14 (100%)
Country of origin		
Italy	10/14 (71%)	14/14 (100%)
Albania	4/14 (29%)	0/14 (0%)
Educational level, N (%)		
Lower secondary	2/14 (14%)	0/14 (0%)
Upper secondary	9/14 (64%)	3/14 (21%)
Higher education	3/14 (21%)	11/14 (79%)
Marital status		
Married	8/14 (57%)	8/14 (57%)
Unmarried		
or cohabitants	6/14 (43%)	6/14 (43%)
Employment status at T0, N (%)		
Employed	13/14 (93%)	14/14 (100%)
Homemaker	1/14 (7%)	0/14 (0%)
Unemployed	0/14 (0%)	0/14 (0%)
Employment status at T2, N (%)		
Employed	13/14 (93%)	14/14 (100%)
Homemaker	1/14 (7%)	0/14 (0%)
Unemployed	0/14 (0%)	0/14 (0%)
Other children		
Yes	8/14 (57%)	8/14 (57%)
No	6/14 (43%)	6/14 (43%)
Children attending day care at T2		
Yes	14/14 (100%)	14/14 (100%)
No	0/14 (0%)	0/14 (0%)

**Table 2 IJNS-11-00014-t002:** Gender, molecular analysis, and biochemical signs at NBS of positive neonates at the NBS.

Positive	Gender	Mutation 1	Mutation 2	GCase Activity μM/h	S-LysoGb1 nmol/L
1	M	p.Asn409Ser	p.Thr408Met	1.42	53
2	M	p.Asn409Ser	p.Asn409Ser	0.60	136
3	M	p.Asn409Ser	p.Leu483Pro	0.44	164
4	F	p.Asn409Ser	P.Gly241Arg	0.38	165
5	M	p.Asn409Ser	p.Asn409Ser	1.26	186
6	M	p.Asn409Ser	p.His294gLN + aSP448His	0.50	232
7	M	p.Asn409Ser	p.Asn409Ser	2.01	NA

GD: Gaucher Diseases; GlcSph: Glucosilsfingosina.

## Data Availability

Data is available on request.

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
