# Peer review of "Newborn Screening for Gaucher Disease: Parental Stress and Psychological Burden"

_2409-515X, 2025, doi:10.3390/ijns11010014_

Round 1

Reviewer 1 Report

Comments and Suggestions for Authors

The authors aim to investigate parental stress in families of newborns with positive results for Gaucher disease through a new born screening (NBS) program in Northeastern Italy. The study evaluates the psychological burden over time on parents of children presymptomatically diagnosed with GD1 via NBS. They highlight the psychological impact of NBS for GD1, emphasizing the need for better, multidisciplinary communication to reduce parental stress throughout the diagnostic and treatment process. Because of the time between diagnosis and disease onset for GD1, the study presumes that parental stress in these cases differs from stress associated with other NBS target diseases. The authors’ proposal for specific measures to mitigate stress is commendable. However, it remains questionable whether all screening centers can realistically implement such comprehensive responses in practice.

Major Points

Lines 109–110

Clarify whether the control group consists of parents of completely healthy children or parents of children with negative NBS results who are nevertheless regularly monitored for certain conditions.

Line 114

Does the T0 PSI assessment occur on the day of diagnosis, or can it be delayed if the baby is 1–2 months old? Parental stress scores may be particularly high on the day of diagnosis, but some parents might report substantially lower scores just a few days later.

Line 199

The explanation in Table 1 indicates a similar educational level between groups. However, there appears to be a significant disparity, with 21% vs. 79% of parents having a higher education level. Additionally, couples who reported during the Short Structured Interview that they did not fully understand the information about NBS or felt it would have been better not to diagnose the condition presymptomatically tended to have a lower education level. This factor likely influenced the study results and warrants further exploration.

Lines 300–301

What specifically accounts for the reduction in PSI scores over time in the G1 group? If the stress decreased due to time alone or other factors, this suggests that improving explanations at the time of diagnosis could similarly reduce PSI scores.

Lines 319–322

Was there a notable difference in PSI scores between the three parents who stated they would have preferred not to have been diagnosed before symptom onset and the remaining 11 who supported early diagnosis? The data suggest that lower education levels are associated with preferring diagnosis after symptoms appear, while higher education levels align with favoring presymptomatic diagnosis.

Table 1

Consider including information on whether parents returned to their hometowns, as well as details on working mothers, such as the duration of childcare leave and whether the child was placed in daycare. These factors may significantly influence parental stress and anxiety. Additionally, for the eight families with G1 siblings, has the PSI of these parents been compared with the PSI of parents of siblings without GD?

Table 2

Were there any premature births or low birth weight infants? Other factors such as weight gain and milk intake could significantly impact parental anxiety and stress levels.

Tables 5–6

How is the "Difficult Child" metric evaluated? For example, while Control Mothers showed little change in this item between T0 and T1 (35⇒38), G1 Mothers exhibited a significant increase (49⇒71). G1 Mothers also showed reductions in "Parental Distress" and "Parent-Child Dysfunction Interaction." Was the child's development assessed as part of the evaluation?

Notation and Formatting Errors

The enzyme activity values in Table 2 are still written in Italian.

The explanation for the table on line 243 references Table 5 but should reference Table 6.

There is no Table 7, and two tables are incorrectly labeled as Table 6.

Author Response

Dear Reviewer,

Thank you for your review of our manuscript entitled "Newborn screening for Gaucher disease: parental stress and psychological burden."

We have revised the article according to your comments; our responses and details of the changes made are outlined below. The changes that have been made within the manuscript file are highlighted in red for your reference.

We await your response and stand ready to address any further questions.

Sincerely,

Prof. Alberto B. Burlina

Comments to the Author

The authors aim to investigate parental stress in families of newborns with positive results for Gaucher disease through a new born screening (NBS) program in Northeastern Italy. The study evaluates the psychological burden over time on parents of children presymptomatically diagnosed with GD1 via NBS. They highlight the psychological impact of NBS for GD1, emphasizing the need for better, multidisciplinary communication to reduce parental stress throughout the diagnostic and treatment process. Because of the time between diagnosis and disease onset for GD1, the study presumes that parental stress in these cases differs from stress associated with other NBS target diseases. The authors’ proposal for specific measures to mitigate stress is commendable. However, it remains questionable whether all screening centers can realistically implement such comprehensive responses in practice.

Major Points

Line 109-110

Clarify whether the control group consists of parents of completely healthy children or parents of children with negative NBS results who are nevertheless regularly monitored for certain conditions.

R: Thank you for your comment.

The control group analyzed in our study consists of parents of newborns were tested at neonatal screening that resulted normal for enzymatic activity. They are healthy newborns that they do not need any futher contols. The sentence "children with negative NBS results" means that they are not affected and they do not need any specific controls.

Line 114

Does the T0 PSI assessment occur on the day of diagnosis, or can it be delayed if the baby is 1–2 months old? Parental stress scores may be particularly high on the day of diagnosis, but some parents might report substantially lower scores just a few days later.

R: Thank you for your comment, which allows us to clarify that T0 has been performed at the moment of diagnosis communication (genetic testing) which allows to define the genotype causing the disease.

Screening process for Gaucher disease provides that the positive screening result is obtained at the 48th hour of the newborn's life, but genetic confirmation of the disease and, consequently, the communication to the family occurs at the first month of life. We agree with the comments that parental stress may be very high before the first month of life.

Our aim was to investigate the level of parental stress related to the communication of diagnosis. We have added the following sentence in the text to better explain: "The communication of the diagnosis occurs with the confirmation of pathologic genetic testing. This happens when the newborn is almost 1 month\ old". (Lines 106-108)

Line 199

The explanation in Table 1 indicates a similar educational level between groups. However, there appears to be a significant disparity, with 21% vs. 79% of parents having a higher education level. Additionally, couples who reported during the Short Structured Interview that they did not fully understand the information about NBS or felt it would have been better not to diagnose the condition presymptomatically tended to have a lower education level. This factor likely influenced the study results and warrants further exploration.

R: Thank you for pointing this out. We agree that there is a disparity in parental educational levels between the groups. To address this, we have included the following clarification and changed the sentence in the text: "Parents in the control group presented higher educational levels, with 79% having higher education, compared to 21% in the GD1 group" (lines 218-219). This addition aims to clarify the observed difference.

We believe that all parents have sufficient cognitive understanding of the information provided by clinicians regarding the disease and its possible clinical presentation. The communication of the diagnosis, facilitated by the presence of a psychologist, is carefully tailored to the specific characteristics and needs of each family.

However, we acknowledge that educational level could indeed influence parental perception.

In response to your comment, we will address this consideration in the limitations section of the manuscript and emphasize the necessity of developing tailored communication strategies. These strategies should ensure that all parents, irrespective of their educational background, receive clear and comprehensible information. This aspect warrants further investigation in future studies to better understand its impact on parental perspectives and decision-making.

We have added the following part in the Limitations section: "An additional limitation is the disparity in educational levels between the control group, which showed a higher level of education, and the GD1 group. This discrepancy may have influenced the perception and acceptance of the NBS result and could be associated with stress levels at the time of diagnosis communication. Future research should account for educational level as a factor that may affect the psychological burden experienced by parents of children with positive NBS results." (lines 454-460).

Lines 300–301

What specifically accounts for the reduction in PSI scores over time in the G1 group? If the stress decreased due to time alone or other factors, this suggests that improving explanations at the time of diagnosis could similarly reduce PSI scores.

R: We thank the reviewer for this important comment. The long-term adjustment to the disease is a complex process involving numerous variables. Primarily, it includes the gradual understanding of the disease, the acquisition of crucial information about it, the realization that the child is showing appropriate psychomotor development, and the positive therapeutic relationship with the care team. These factors assist parents in the coping process and in acquiring strategies that support adaptation.

Additionally, unlike other metabolic diseases at risk of acute decompensation, patients with Gaucher disease do not present with acute symptomatic episodes and therefore do not require long hospitalizations. Certainly, the quality of the diagnosis communication is also a key element in building an adequate adaptation process to the disease. In the discussion section, we reported that establishing a standardized communication practice for newborn screening and implementing patient- and family-centered care is essential for chronic diseases in pediatric patients.

Further studies are needed to analyze parental stress in the long term, particularly in metabolic diseases at risk of acute decompensation. We have added this consideration in the conclusion section: "Further studies are needed to analyze long-term parental stress in metabolic diseases at risk of acute decompensation." (Lines 471-472).

Lines 319–322

Was there a notable difference in PSI scores between the three parents who stated they would have preferred not to have been diagnosed before symptom onset and the remaining 11 who supported early diagnosis? The data suggest that lower education levels are associated with preferring diagnosis after symptoms appear, while higher education levels align with favoring presymptomatic diagnosis.

R: Thank you for the comment. We agree with the importance of considering the educational level of parents in evaluating their perception. However, the small size of our sample does not allow for more targeted statistical analyses.

Future studies will need to consider this important aspect, and we have included this point in the Limitations section: "Future research should account for educational level as a factor that may affect the psychological burden experienced by parents of children with positive NBS results." (lines 458-460).

Table 1

Consider including information on whether parents returned to their hometowns, as well as details on working mothers, such as the duration of childcare leave and whether the child was placed in daycare. These factors may significantly influence parental stress and anxiety. Additionally, for the eight families with G1 siblings, has the PSI of these parents been compared with the PSI of parents of siblings without GD?

R: Thank you for the comment; this suggestion is very helpful. We have added the following information in Table 1: "Country of origin, Employment status at T2, Children attending day care at T2" (Line 220).

Regarding the comparison with healthy siblings, although it would have been very useful at the time of data collection, it was not included. However, it is a suggestion we are considering for future studies to better understand parental stress in parents of children with Gaucher disease.

Table 2

Were there any premature births or low birth weight infants? Other factors such as weight gain and milk intake could significantly impact parental anxiety and stress levels.

R: Thank you for the comment. There were no pre-term or clinically low-weight newborns in our study. We have added the following information to the Study cohort section to explicitly state this: "All the GD1 and control newborns considered in the study were full-term, with a weight comparable to healthy peers of the same age. No infant, at the time of birth or during the 36-month follow-up, showed any clinical signs of the disease." (Lines 203-205).

Tables 5–6

How is the "Difficult Child" metric evaluated? For example, while Control Mothers showed little change in this item between T0 and T1 (3538), G1 Mothers exhibited a significant increase (4971). G1 Mothers also showed reductions in "Parental Distress" and "Parent-Child Dysfunction Interaction." Was the child's development assessed as part of the evaluation?

R: Thank you for the comment. In this study, we utilized the Parenting Stress Index, a questionnaire specifically designed to assess parenting stress. This refers to the particular stress perceived by parents in relation to their child, themselves as parents, and the overall family system. One of the three scales comprising the total parenting stress score is the "Difficult Child" scale, which focuses on the parent's perception of fundamental characteristics of the child's behavior that make parenting the child easier or more challenging.

As reported in the questionnaire manual guidelines, this scale was calculated by summing the scores of the related items (items 25–36), which contribute to the raw score of the scale. The "Difficult Child" raw score was then converted into percentiles based on the child's age using normative tables provided in the manual. This scale aims to assess the parents' difficulties related to the management of their child's behavior.

Regarding child development assessment, we evaluated psychomotor development using dedicated developmental scales as part of our clinical practice. We did not report the scores here because all the newborns showed normal development. We have added this information in the text: "Psychomotor development, assessed with a standardized developmental scale, was normal for all GD1 children." (lines 205-207).

Notation and Formatting Errors

The enzyme activity values in Table 2 are still written in Italian.

R: Thank you for the comment. We have made the correction.

The explanation for the table on line 243 references Table 5 but should reference Table 6.

R: Thank you for the comment. We have made the correction.

There is no Table 7, and two tables are incorrectly labeled as Table 6.

R: Thank you for the comment. We have made the correction.

Comments to the Author

The authors aim to investigate parental stress in families of newborns with positive results for Gaucher disease through a new born screening (NBS) program in Northeastern Italy. The study evaluates the psychological burden over time on parents of children presymptomatically diagnosed with GD1 via NBS. They highlight the psychological impact of NBS for GD1, emphasizing the need for better, multidisciplinary communication to reduce parental stress throughout the diagnostic and treatment process. Because of the time between diagnosis and disease onset for GD1, the study presumes that parental stress in these cases differs from stress associated with other NBS target diseases. The authors’ proposal for specific measures to mitigate stress is commendable. However, it remains questionable whether all screening centers can realistically implement such comprehensive responses in practice.

Major Points

Line 109-110

Clarify whether the control group consists of parents of completely healthy children or parents of children with negative NBS results who are nevertheless regularly monitored for certain conditions.

R: Thank you for your comment.

The control group analyzed in our study consists of parents of newborns were tested at neonatal screening that resulted normal for enzymatic activity. They are healthy newborns that they do not need any futher contols. The sentence "children with negative NBS results" means that they are not affected and they do not need any specific controls.

Line 114

Does the T0 PSI assessment occur on the day of diagnosis, or can it be delayed if the baby is 1–2 months old? Parental stress scores may be particularly high on the day of diagnosis, but some parents might report substantially lower scores just a few days later.

R: Thank you for your comment, which allows us to clarify that T0 has been performed at the moment of diagnosis communication (genetic testing) which allows to define the genotype causing the disease.

Screening process for Gaucher disease provides that the positive screening result is obtained at the 48th hour of the newborn's life, but genetic confirmation of the disease and, consequently, the communication to the family occurs at the first month of life. We agree with the comments that parental stress may be very high before the first month of life.

Our aim was to investigate the level of parental stress related to the communication of diagnosis. We have added the following sentence in the text to better explain: "The communication of the diagnosis occurs with the confirmation of pathologic genetic testing. This happens when the newborn is almost 1 month\ old". (Lines 106-108)

Line 199

The explanation in Table 1 indicates a similar educational level between groups. However, there appears to be a significant disparity, with 21% vs. 79% of parents having a higher education level. Additionally, couples who reported during the Short Structured Interview that they did not fully understand the information about NBS or felt it would have been better not to diagnose the condition presymptomatically tended to have a lower education level. This factor likely influenced the study results and warrants further exploration.

R: Thank you for pointing this out. We agree that there is a disparity in parental educational levels between the groups. To address this, we have included the following clarification and changed the sentence in the text: "Parents in the control group presented higher educational levels, with 79% having higher education, compared to 21% in the GD1 group" (lines 218-219). This addition aims to clarify the observed difference.

We believe that all parents have sufficient cognitive understanding of the information provided by clinicians regarding the disease and its possible clinical presentation. The communication of the diagnosis, facilitated by the presence of a psychologist, is carefully tailored to the specific characteristics and needs of each family.

However, we acknowledge that educational level could indeed influence parental perception.

In response to your comment, we will address this consideration in the limitations section of the manuscript and emphasize the necessity of developing tailored communication strategies. These strategies should ensure that all parents, irrespective of their educational background, receive clear and comprehensible information. This aspect warrants further investigation in future studies to better understand its impact on parental perspectives and decision-making.

We have added the following part in the Limitations section: "An additional limitation is the disparity in educational levels between the control group, which showed a higher level of education, and the GD1 group. This discrepancy may have influenced the perception and acceptance of the NBS result and could be associated with stress levels at the time of diagnosis communication. Future research should account for educational level as a factor that may affect the psychological burden experienced by parents of children with positive NBS results." (lines 454-460).

Lines 300–301

What specifically accounts for the reduction in PSI scores over time in the G1 group? If the stress decreased due to time alone or other factors, this suggests that improving explanations at the time of diagnosis could similarly reduce PSI scores.

R: We thank the reviewer for this important comment. The long-term adjustment to the disease is a complex process involving numerous variables. Primarily, it includes the gradual understanding of the disease, the acquisition of crucial information about it, the realization that the child is showing appropriate psychomotor development, and the positive therapeutic relationship with the care team. These factors assist parents in the coping process and in acquiring strategies that support adaptation.

Additionally, unlike other metabolic diseases at risk of acute decompensation, patients with Gaucher disease do not present with acute symptomatic episodes and therefore do not require long hospitalizations. Certainly, the quality of the diagnosis communication is also a key element in building an adequate adaptation process to the disease. In the discussion section, we reported that establishing a standardized communication practice for newborn screening and implementing patient- and family-centered care is essential for chronic diseases in pediatric patients.

Further studies are needed to analyze parental stress in the long term, particularly in metabolic diseases at risk of acute decompensation. We have added this consideration in the conclusion section: "Further studies are needed to analyze long-term parental stress in metabolic diseases at risk of acute decompensation." (Lines 471-472).

Lines 319–322

Was there a notable difference in PSI scores between the three parents who stated they would have preferred not to have been diagnosed before symptom onset and the remaining 11 who supported early diagnosis? The data suggest that lower education levels are associated with preferring diagnosis after symptoms appear, while higher education levels align with favoring presymptomatic diagnosis.

R: Thank you for the comment. We agree with the importance of considering the educational level of parents in evaluating their perception. However, the small size of our sample does not allow for more targeted statistical analyses.

Future studies will need to consider this important aspect, and we have included this point in the Limitations section: "Future research should account for educational level as a factor that may affect the psychological burden experienced by parents of children with positive NBS results." (lines 458-460).

Table 1

Consider including information on whether parents returned to their hometowns, as well as details on working mothers, such as the duration of childcare leave and whether the child was placed in daycare. These factors may significantly influence parental stress and anxiety. Additionally, for the eight families with G1 siblings, has the PSI of these parents been compared with the PSI of parents of siblings without GD?

R: Thank you for the comment; this suggestion is very helpful. We have added the following information in Table 1: "Country of origin, Employment status at T2, Children attending day care at T2" (Line 220).

Regarding the comparison with healthy siblings, although it would have been very useful at the time of data collection, it was not included. However, it is a suggestion we are considering for future studies to better understand parental stress in parents of children with Gaucher disease.

Table 2

Were there any premature births or low birth weight infants? Other factors such as weight gain and milk intake could significantly impact parental anxiety and stress levels.

R: Thank you for the comment. There were no pre-term or clinically low-weight newborns in our study. We have added the following information to the Study cohort section to explicitly state this: "All the GD1 and control newborns considered in the study were full-term, with a weight comparable to healthy peers of the same age. No infant, at the time of birth or during the 36-month follow-up, showed any clinical signs of the disease." (Lines 203-205).

Tables 5–6

How is the "Difficult Child" metric evaluated? For example, while Control Mothers showed little change in this item between T0 and T1 (3538), G1 Mothers exhibited a significant increase (4971). G1 Mothers also showed reductions in "Parental Distress" and "Parent-Child Dysfunction Interaction." Was the child's development assessed as part of the evaluation?

R: Thank you for the comment. In this study, we utilized the Parenting Stress Index, a questionnaire specifically designed to assess parenting stress. This refers to the particular stress perceived by parents in relation to their child, themselves as parents, and the overall family system. One of the three scales comprising the total parenting stress score is the "Difficult Child" scale, which focuses on the parent's perception of fundamental characteristics of the child's behavior that make parenting the child easier or more challenging.

As reported in the questionnaire manual guidelines, this scale was calculated by summing the scores of the related items (items 25–36), which contribute to the raw score of the scale. The "Difficult Child" raw score was then converted into percentiles based on the child's age using normative tables provided in the manual. This scale aims to assess the parents' difficulties related to the management of their child's behavior.

Regarding child development assessment, we evaluated psychomotor development using dedicated developmental scales as part of our clinical practice. We did not report the scores here because all the newborns showed normal development. We have added this information in the text: "Psychomotor development, assessed with a standardized developmental scale, was normal for all GD1 children." (lines 205-207).

Notation and Formatting Errors

The enzyme activity values in Table 2 are still written in Italian.

R: Thank you for the comment. We have made the correction.

The explanation for the table on line 243 references Table 5 but should reference Table 6.

R: Thank you for the comment. We have made the correction.

There is no Table 7, and two tables are incorrectly labeled as Table 6.

R: Thank you for the comment. We have made the correction.

Reviewer 2 Report

Comments and Suggestions for Authors

The study included only 7 families with children diagnosed with Gaucher Disease Type 1 (GD1) through newborn screening. This small sample size limits the generalizability of the findings.

The study excluded 4 families due to language barriers and 3 due to refusal to participate. This could introduce selection bias if these excluded families differed systematically from those included.

While the study assessed parental stress at 12 and 36 months, longer-term follow-up would be valuable to understand the ongoing psychological impact as children age.

The control group consisted of parents whose children had negative newborn screening results. A more appropriate comparison may have been parents of children diagnosed with GD1 through clinical presentation rather than screening.I urge the authors to include that group  if passible.

The study does not appear to control for other factors that could influence parental stress, such as socioeconomic status, family support systems, or parents' pre-existing mental health conditions.

The study primarily uses self-reported questionnaires, which can be subject to recall bias and social desirability bias.

While the study included both mothers and fathers, there seems to be less in-depth analysis of fathers' experiences and stress levels.

The study could have benefited from more qualitative data to provide deeper insights into parents' experiences and coping strategies.

The study was conducted in Northeastern Italy, and cultural factors may influence parental stress and coping. This limits the generalizability of findings to other cultural contexts.

The study raises but does not fully address the ethical implications of newborn screening for conditions like GD1, where treatment may not be initiated immediately.

The limitations should be clearly written in the discussion section.

I am curious to know how these patients were followed up:

-How frequently were follow-up appointments conducted?

-What specific tests were performed during these follow-ups?

-Were the patients seen by the same healthcare provider consistently, or did they see different providers?

All of these factors might have an impact on parental stress. It would be helpful if the authors mentioned their follow-up approach for these cases in detail.

Despite these limitations, the study provides valuable insights into the psychological impact of newborn screening for Gaucher Disease on parents and highlights the need for improved communication and support strategies in newborn screening programs.

Author Response

Dear Reviewer,

Thank you for your review of our manuscript entitled "Newborn screening for Gaucher disease: parental stress and psychological burden."

We have revised the article according to your comments; our responses and details of the changes made are outlined below. The changes that have been made within the manuscript file are highlighted in red for your reference.

We await your response and stand ready to address any further questions.

Sincerely,

Prof. Alberto B. Burlina

The study included only 7 families with children diagnosed with Gaucher Disease Type 1 (GD1) through newborn screening. This small sample size limits the generalizability of the findings.

R: Thank you for pointing this out. The small sample size may limit the generalizability of our results. However, GD1 is a rare metabolic disease with a prevalence of 1:60,000–1:100,000 (Orphanet, 2025), making it challenging to recruit a large number of families. Nevertheless, this limitation may restrict the possibility to generalize our results; this information has been added to the Limitations paragraph:

"The small sample size and the variables related to sociocultural aspects and the region of origin of the parents, may have affected the results. Further studies are needed to overcome these limitations and to produce more generalizable results." (lines 447-449).

The study excluded 4 families due to language barriers and 3 due to refusal to participate. This could introduce selection bias if these excluded families differed systematically from those included.

R: Thank you for your consideration. We acknowledge that the exclusion of some families may have introduced bias. However, our primary aim was to investigate, using interviews as well, the specific perspectives of mothers and fathers of newborns who screened positive for GD1. Therefore, some parents were excluded due to significant language barriers. We have added this aspect to the Limitations paragraph: “Moreover, in our study we did not consider parents due to language barriers and parents who refused to participate, so these aspects could represent a selection bias. Future studies should aim to include all families, using instruments created in different languages." (lines 450-453)

While the study assessed parental stress at 12 and 36 months, longer-term follow-up would be valuable to understand the ongoing psychological impact as children age.

R: Thank for the comment. One of our main objectives will certainly be to continue the follow-up of these families in the long term, to evaluate how the adaptation process of parents progresses several years after the communication of the diagnosis. Given the importance of the aspect you highlighted, we have included this point in the Future Directions section: “Studies with longer-term follow-up are needed to evaluate how the adaptation process of parents of children with GD1 evolves over time(lines 473-474).

The control group consisted of parents whose children had negative newborn screening results. A more appropriate comparison may have been parents of children diagnosed with GD1 through clinical presentation rather than screening. I urge the authors to include that group if passible.

R: We thank the reviewer for this comment. Unfortunately, there are no GD1 patients presenting with clinical symptomatology during the neonatal period. Therefore, this type of comparison is not feasible.

The study does not appear to control for other factors that could influence parental stress, such as socioeconomic status, family support systems, or parents' pre-existing mental health conditions.

R: Thank you for the consideration. We controlled for key socioeconomic and family characteristics and found that the groups were comparable in most aspects. Additionally, we have included further details about the groups in Table 1 to provide a clearer explanation of their characteristics: "Country of origin, Employment status at T2, Children attending day care at T2" (line 220).

A notable difference emerged only in educational level, which we have modified in the text (lines 218–219) and acknowledged in the limitations section (lines 454-457).

The study primarily uses self-reported questionnaires, which can be subject to recall bias and social desirability bias.

R: Thank you for your comment. We decided to use a self-report instrument to quantitatively assess parents' subjective perception of their stress. However, self-report instruments can be susceptible to social desirability or defensive bias. To reduce the risk of these biases, we selected the Parenting Stress Index (PSI) questionnaire, which includes a "Defensive Responding Scale" designed to control for the risk of defensive responses related to social desirability. All participants obtained non-clinical scores on this scale.

While the study included both mothers and fathers, there seems to be less in-depth analysis of fathers' experiences and stress levels.

R: Thank you for the comment. We are very interested to analyze the perspective of both mothers and also fathers, who are typically not included in research. We decided to implement the same analyses for all parents, but specific focus on the role of fathers is a very interesting topic to investigate in future studies. We agree with your suggestion, we added a part in the Future studies paragraph (lines 474-477).

The study could have benefited from more qualitative data to provide deeper insights into parents' experiences and coping strategies.

R: Thank you for the consideration. We opted to use self-report questionnaires to quantitatively capture both mothers' and fathers' perceptions regarding the experience of receiving positive NBS results for GD1. We agree that future studies could implement this approach with more in-depth interviews to qualitatively explore parental perspectives. Additionally, more specific psychological aspects, including parental coping strategies and their associations with parenting stress, may be assessed. This would allow for a more nuanced understanding, including potential differences in the emotions and opinions of fathers compared to mothers.

The study was conducted in Northeastern Italy, and cultural factors may influence parental stress and coping. This limits the generalizability of findings to other cultural contexts.

R: Thank you for the comment. The study was conducted in Northeastern Italy, but the families with a child with positive NBS result for GD1 originated from various regions of Italy, and some were from Eastern Europe. This information has been added to Table 1 (line 220). Nevertheless, cultural factors may limit the generalizability of our results; this information has been added to the Limitations paragraph: "The small sample size and the variables related to sociocultural aspects and the region of origin of the parents, may have affected the results. Further studies are needed to overcome these limitations and to produce more generalizable results." (lines 447-449).

The study raises but does not fully address the ethical implications of newborn screening for conditions like GD1, where treatment may not be initiated immediately.

R: Thank you for this very important comment.

NBS is an important and fundamental prevention process for early identification of pathological conditions and timely intervention before the onset of symptoms in acute-onset diseases at risk of decompensation. For Gaucher disease type I, this aspect remains controversial because GD1 it is not an early-onset disease, and its onset can be unpredictable, with symptoms varying considerably from patient to patient. Moreover, a positive NBS result for GD1 is found in newborns who do not present any symptomatic manifestations. This aspect makes the communication of the diagnosis even more complex and makes the global and multidisciplinary care of the newborn and their family fundamental to support parents towards a sufficiently good quality of life. Some authors reported that the patients with GD1 considering in their study, in the initially untreated cohort did not require any treatment ; these authors supports an a watch-and-wait approach rather than early start of GD1 specific therapy for asymptomatic or mildly symptomatic patients. We add the following consideration in the discussion section as request: “Newborns positive for GD1 are asymptomatic at birth, and the age at onset and clinical presentation of Type I GD is variable [28]. Furthermore, the onset is unpredictable and can potentially occur at any time from infancy to adulthood. Some authors support a watch-and-wait approach rather than early start of GD1-specific therapy for asymptomatic or mildly symptomatic patients [29]. Therefore, it is desirable to think that treatment should begin upon the presentation of signs or symptoms related to the disease. Additionally, it is possible that some newborns with positive NBS results may never develop symptoms [29]. These considerations raise an important discussion regarding the ethical aspects related to the early identification of this disease through NBS. Therefore diagnosis communication may be even more complex and underscores the importance of global and multidisciplinary care for the newborn and their family to support parents in maintaining a sufficiently good quality of life. Therefore, a comprehensive approach to care and support for these families is essential." (lines 375-388).

The limitations should be clearly written in the discussion section.

R: Thank you for the comments. We have added a specific section on the limitations of the study (Limitations section), in which we have included the elements that may represent important limitations in our study.

I am curious to know how these patients were followed up:

-How frequently were follow-up appointments conducted?

R. Thank you for your comment.

Outpatients visits were performed every 6 month during the first year of life and annually in the following years. We have found this approach to be very useful both for monitoring clinical aspects related to the onset of symptoms and for maintaining a relationship with the parents, providing them with a space to clarify doubts and answer their questions. We add this part in the study cohort section: “Outpatients visits were performed every 6 months during the first year of life and annually in the following years (Lines 208-209).

-What specific tests were performed during these follow-ups?

R: Thank you for the question.

Follow-up, included biochemical tests and abdominal ultrasound. Additionally, specific assessment was performed to define and monitor over time the acquisition of psychomotor development milestones through the administration of psychometric and age-related developmental scale(i.e. Bayley-III, Griffiths or Wechsler Scale). We add this part in the study cohort section: The follow-up visits included biochemical tests, abdominal ultrasound, and specific assessments to define and monitor the acquisition of psychomotor development milestones over time through the administration of age-appropriate psychometric and developmental scales (Lines 209-212).

-Were the patients seen by the same healthcare provider consistently, or did they see different providers?

R:Thank you for the comment. At each follow-up visit, the parents always meet with the same healthcare professionals, including the pediatrician specialized in inherited metabolic diseases and the same psychologist. The patient’s care by a multidisciplinary team with consistent reference figures is very useful in establishing an adequate therapeutic relationship between the parental couple and the care team. We add this part in the discussion section: “Considering the psychological impact of a disease diagnosis, it is crucial to establish a follow-up approach with parents of newborns affected by GD1 that is adequate not only for monitoring the onset of symptoms but also for providing parents with a space for listening and emotional processing. This is essential for building an appropriate therapeutic alliance. Our approach ensures that parents see the same professional figures (metabolic pediatrician and psychologist) at each follow-up visit, allowing them to benefit from continuity of care and appropriate support" (Lines 436-442).

All of these factors might have an impact on parental stress. It would be helpful if the authors mentioned their follow-up approach for these cases in detail.

R: Thank you for the comments. We add the follow-up approach in the Results section and in the Discussion section.

Despite these limitations, the study provides valuable insights into the psychological impact of newborn screening for Gaucher Disease on parents and highlights the need for improved communication and support strategies in newborn screening programs.

Round 2

Reviewer 1 Report

Comments and Suggestions for Authors

Although the statement has been updated to indicate that the control group had a higher educational level than the GD1 group, the text on line 213 remains unchanged: "The groups were similar in terms of age, ethnicity, educational level, marital status, and professional status." Notably, the phrase "educational level" has not been removed, creating an inconsistency with the revised statement.

Author Response

Dear Reviewer

Thank you for your review of our manuscript entitled "Newborn screening for Gaucher disease: parental stress and psychological burden."

We have revised the article according to your comments.

We await your response and stand ready to address any further questions.

Sincerely,

Prof. Alberto B. Burlina

Although the statement has been updated to indicate that the control group had a higher educational level than the GD1 group, the text on line 213 remains unchanged: "The groups were similar in terms of age, ethnicity, educational level, marital status, and professional status." Notably, the phrase "educational level" has not been removed, creating an inconsistency with the revised statement.

R: Thank you for your comment. We have removed the phrase "educational level" (line 217), as indicated. Unfortunately, there was a typographical error in the text. We have kept the sentence "Parents in the control group presented higher educational levels, with 79% having higher education, compared to 21% in the GD1 group." (lines 218-219).